# Risk Assessment of New Energy Vehicle Supply Chain Based on Variable Weight Theory and Cloud Model: A Case Study in China

**Qingyou Yan [1,2], Meijuan Zhang [1,2,\*], Wei Li [1,2] and Guangyu Qin [1,2,\*]** 

1   School of Economic & Management, North China Electric Power University, Beijing 102206, China; yanqingyou@ncepu.edu.cn (Q.Y.); liwei502520@163.com (W.L.)
2   Beijing Key Laboratory of New Energy & Low Carbon Development, North China Electric Power University, Beijing 102206, China
\*   Correspondence: zhangmeijuan@ncepu.edu.cn (M.Z.); qinguangyu@ncepu.edu.cn (G.Q.)

**Abstract:** In order to protect the environment and reduce energy consumption, new energy vehicles have begun to be vigorously promoted in various countries. In recent years, the rise of intelligent technology has had a great impact on the supply chain of new energy vehicles, which, coupled with the complexity of the supply chain itself, puts it at great risk. Therefore, it is quite indispensable to evaluate the risk of the new energy vehicle supply chain. This paper assesses the risks faced by China's new energy vehicle supply chain in this period of technological transformation. First of all, this paper establishes an evaluation criteria system of 16 sub-criterion related to three dimensions: the market risk, operational risk, and the environmental risk. Then, variable weight theory is proposed to modify the constant weight obtained by the fuzzy analytic hierarchy process (FAHP). Finally, a risk assessment of China's new energy vehicle supply chain is carried out by combining the variable weight and the cloud model. This method can effectively explain the randomness of matters, and avoid the influence of value abnormality on the criteria system. The results show that China's new energy vehicle supply chain is at a high level. Through the identification of risk factors, mainly referring to the low clustering risk, technical level risk and information transparency risk, this paper can provide a risk prevention reference for corresponding enterprises.

**Keywords:** new energy vehicles; supply chain risk assessment; variable weight theory; cloud model; fuzzy analytic hierarchy process

---

## 1. Introduction

Climate change is a global environmental challenge closely related to economic growth and social development. The cost of rapid economic development is excessive energy consumption and environmental pollution. At present, how to realize energy conservation and emission reduction are growing concerns around the world. In the Paris Agreement, China promised to achieve the goal of peak carbon dioxide emissions by 2030 [1], and this was reflected in the 13th Five-Year Plan. The automobile industry is the main driver of energy consumption and carbon emissions [2]. Therefore, during the 13th Five-Year Plan period, China vigorously promoted new energy vehicles (NEV) and directed the electric transformation of the traditional automobile industry [3]. During the following 14th Five-Year Plan period, the development of new energy vehicles will continue to be included in China's strategic measures to address climate change and promote green development [4]. Other countries have also taken measures to reduce environmental pollution in the automobile industry. For example, the European Union issued the "2019/631 document" in 2019, stipulating that the carbon dioxide

emissions of newly registered passenger vehicles in 2025 and 2030 should be reduced by 15% and 37.5%, respectively, on the basis of 2021, and the development of new energy vehicles has become the only choice; the United States has also adopted tax relief on the demand side and an accumulated points system on the supply side to promote new energy vehicles to cope with environmental changes [5].

Thus, the research on new energy vehicles mainly focuses on technology [6–8], operation mode [9–12], environmental protection assessment [13–15], market demand analysis [16,17], and so on. For example, Perera et al. proposed a charging infrastructure planning model in the urban environment. The case study shows that the model can select the best capacity and location of electric vehicles, maximize the coverage service, and minimize the life cycle cost [18]. Lin et al. investigated the positive significance of electric vehicle policy on the environment by establishing models to conduct scenario analyses [7]. Okada and many other scholars analyzed the market demand of new energy vehicles from various perspectives, including perception, personality, and environmental awareness, etc. [19–21]. Nowadays, electrification, networking, intelligence, and sharing are becoming the development currents of automobile industry, which will lead to enormous changes in the structure of the supply chain and suppliers. More and more voices are calling for the establishment of an efficient and high-quality supply chain for new energy vehicles. However, in virtue of its bulkiness and complexity, it faces many risks in the transformation and innovation stages. Consequently, there is a tremendous necessity to assess the risks faced by the supply chain of new energy vehicles in this period; yet there are relatively few studies on the risk assessment of the NEV supply chain [22,23]. Most studies only focus on a single risk factor, and a comprehensive and exclusive assessment of NEV supply chain risk in the transition period is rarely involved. Li et al. studied the influence of subsidy policy on the production decision-making of new energy vehicle manufacturers, and concluded that for new energy vehicle manufacturers considering battery recovery, battery recovery rate, and subsidy policy have a key impact on their competitive position [24]. Kalaitzi et al. discussed the relationship between and the influence of supply chain enterprises on the production process of electric vehicles through a specific case analysis [25]. Only Wu et al. used the fuzzy synthetic evaluation method to evaluate the risk of China's electric vehicle supply chain, and found that for the moment, the risk associated with China's electric vehicle supply chain mainly comes from technical risks and market risks [26]. On the other hand, in terms of the methods of risk assessment, scholars generally adopt analytic hierarchy process (AHP) [27], technique for order preference by similarity to ideal solution (TOPSIS) [28], and fuzzy comprehensive evaluation [26,29]. Mihalis and Thanos applied the failure mode and effect analysis technique (FMEA) to identify and assess the sustainability-related risks of supply chain, and tested the potential correlation between the identified risks, and then gave reasonable suggestions [30]. Song et al. proposed a rough weighted decision-making and trial evaluation laboratory (DEMATEL) to identify critical risk factors in sustainable supply chain management [31].

For the purpose of exploring the risk of China's NEV supply chain, this paper establishes an assessment criteria system; the evaluation results were obtained through a combination of the fuzzy AHP modified by variable weight theory and cloud model, so as to provide reference for relevant enterprises to formulate corresponding risk management strategies and avoidance measures. This paper makes some improvements in the following aspects. Firstly, the new criteria, low clustering risk and low responsiveness risk, fit into the evaluation criteria system for the first time. Secondly, the constant weight obtained by fuzzy AHP is modified using variable weight theory in the process of weighting; the variable weight can reflect the influence of individual criterion outliers. Thirdly, this paper introduces the cloud model for evaluation initiatively.

The rest of the sections are organized as follows. Section 2 is related literature review and Section 3 is the definition of the assessment criteria system. Then, the relevant methodology is presented in Section 4 and the analysis process and results are carried out in Section 5. Finally, Section 6 presents the conclusions.

## 2. Literature Review

This section focuses on the theoretical background related to the NEV supply chain and supply chain risk management, which lays a solid theoretical foundation for the subsequent supply chain risk assessment of new energy vehicles.

### 2.1. New Energy Vehicle Supply Chain

Since its inception, supply chain management has received extensive attention. It has become a relatively prominent research field in the development of decades, and gradually becomes a melting pot of multiple disciplines [32]. Mentzer et al. defined the supply chain as a whole structure that multiple entities directly participate in the information flow, material flow, capital flow and services between the upstream and downstream. This upstream and downstream process starts from the source of raw materials to the final end consumers [33]. Under this definition, supply chain management focuses on the strategic and systematic coordination of various business functions in the supply chain to improve the long-term performance of the whole supply chain. Yang and He pointed out that the basic characteristics of the NEV supply chain are as follows: a) the exploration time is short, and it has high vulnerability; (2) the terminal demand of the supply chain is difficult to predict; (3) it is highly dependent on technology; once a technical link fails, the supply chain management cannot continue [34]. Masiero et al. also raised that the characteristics of the electric vehicle (EV) value chain are different from those of the traditional auto industry. The high complexity of integration and the high cost of components led to the fact that the global integrated EV value chain has not been formed yet, and the concept of the value chain is the substitute of the supply chain in the field of strategic management [35].

The existing research on the supply chain of new energy vehicles mainly focuses on the end consumer side, such as consumer preference [36], market acceptance [37] and market strategy [38]. Noori and Tatar studied the prediction of the new energy regional market, and built an agent-based model to explore the penetration rate of NEV in the United States, explaining the impact of word-of-mouth effect and government subsidies on market development [39]. Huang et al. analyzed the impact of government subsidies on the NEV supply chain in the duopoly market where the NEV supply chain and the traditional vehicle supply chain coexist [40]. Furthermore, Luo et al. only studied the supply chain of NEV [41]. Günther et al. researched the overall transformation process of the supply chain of the automobile industry in the next 20 years through a comprehensive linear optimization model based on real data and scenarios. This process addressed the electric vehicles in Europe and China, and evaluated the challenges of electric vehicles in the future development scenarios [42]. These literatures provide theoretical guidance for the study, but seldom involve in risk management. This paper proposes to make up for the above vacancy by visual model to evaluate the risk of NEV supply chain in the period of electrification transformation.

### 2.2. Supply Chain Risk Management

Supply chain risk (SCR) is an extension of the company's internal risk management thought, which has been widely concerned in recent years. Many scholars have defined supply chain risk from the perspective of supply risk [43–45]. Prakash et al. defined SCR as the possible events in the supply chain that lead to financial loss of the company [46]. Zsidisin et al. defined SCR as the probability of occurrence of an accident due to untimely supply, which will lead to consequences that the affected company cannot cope with, such as inability to meet demand, etc. [47]. This study adopts Zsidisin's theory of SCR. Although the specific definitions are various, scholars still focus on the classification of triggering events, which is understood as the starting point of supply chain risk management [30]. Supply chain risk management (SCRM) is a comprehensive process to identify, analyze, accept or mitigate SCRs [48]. SCR identification is to classify and analyze various potential risks. As mentioned before, as the starting point of risk identification, risks need to be classified

according to different sources, such as according to the SCOR model process or simply divided into internal and non-internal risks [49,50]. Moreover, in order to protect the environment and reduce energy waste, SCRM increasingly needs to consider sustainability and green requirements [27,51]. Different research objects and periods will lead to various SCRs' assessment criterion. For example, Muhammad et al. divided the SCRs of the automobile industry into six categories: external risks, industry risks, organizational risks, operational risks, supply-side risks and demand-side risks [51]. Teresa et al. built an inbound supply risk classification system based upon four categories, namely internal controllable, internal partially controllable, internal uncontrollable and external controllable [52]. On the basis of related literature, this paper constructs the risk assessment system of NEV according to the requirements in the transformation period, particularly in adding low clustering risk and low responsiveness risk.

SCR modeling approaches can be roughly divided into qualitative, quantitative and hybrid models. In the risk analysis and evaluation stage, qualitative or hybrid models are often used, such as FMEA, AHP and TOPSIS; and quantitative technology is mostly used to simulate optimization, so as to make strategy selection [52–54]. Anna et al. developed integrated risk identification and analysis methods by incorporating widely used SCRM tools. Specifically, RBS framework is used to guide risk identification and avoid possible blind spots, while RBM is employed to investigate the occurrence of risk, and then KPI is applied for effect assessment [49]. Mangla et al. applied the fuzzy AHP approach to analyze the risks identified in the green supply chain [27]. Based on the above, this paper further optimizes the application method, in one way by modifying fuzzy AHP by variable weight theory, and firstly introducing cloud model for analysis.

## 3. Determination of NEV Supply Chain Risk Assessment Criteria System

To better carry out the NEV supply chain risk assessment, the establishment of an assessment criteria system is of great importance. The literature, which is extensive, on risk factors associated with the NEV supply chain was studied, and it was found that a large number of factors threaten the robustness of the supply chain [22,26,28]. Meanwhile, we also collected the views of five experts from the fields of energy, new energy vehicles, supply chain management, etc. Through a literature review and expert consultation, this paper divides the risk factors of the NEV supply chain into three dimensions: market risk, operational risk, and environment risk. The specific risk assessment criteria system is shown in Figure 1.

### 3.1. Market Risk (E1)

Market risk refers to the direct and indirect impact of uncertain market factors on various links in the supply chain. This paper considers the risk factors that may cause damage to the supply chain in terms of supply, demand, and the market [22,28,55,56].

### 3.1.1. Supplier Selection Risk (E11)

The development of information technology makes the NEV supply chain network worldwide, thus unreasonable supplier selection will cause a domino effect in the subsequent supply chain. Supplier selection risk is the threat that the quantity and quality of selected suppliers may cause to the supply chain. Too many suppliers will result in increased procurement costs and redundant operations. Conversely, too few suppliers will threaten their stability. Supplier quality involves factors that may produce ventures, such as production technology and flexibility [28].

### 3.1.2. Delivery Delay or Substandard Product Quality Risk (E12)

Lean production is a countermeasure for NEV production in terms of meeting the requirements of the intelligent era. Delay in supplier delivery time will generate a ripple effect on the subsequent flow of the NEV production line. Failure to meet just-in-time rules or non-compliance with product quality will be fatal to the product life cycle. Falling short of the expected production time and quality

standards will not only cause damage to core companies, but also affect the brand reputation of new energy vehicles dominated by the core enterprises [34].

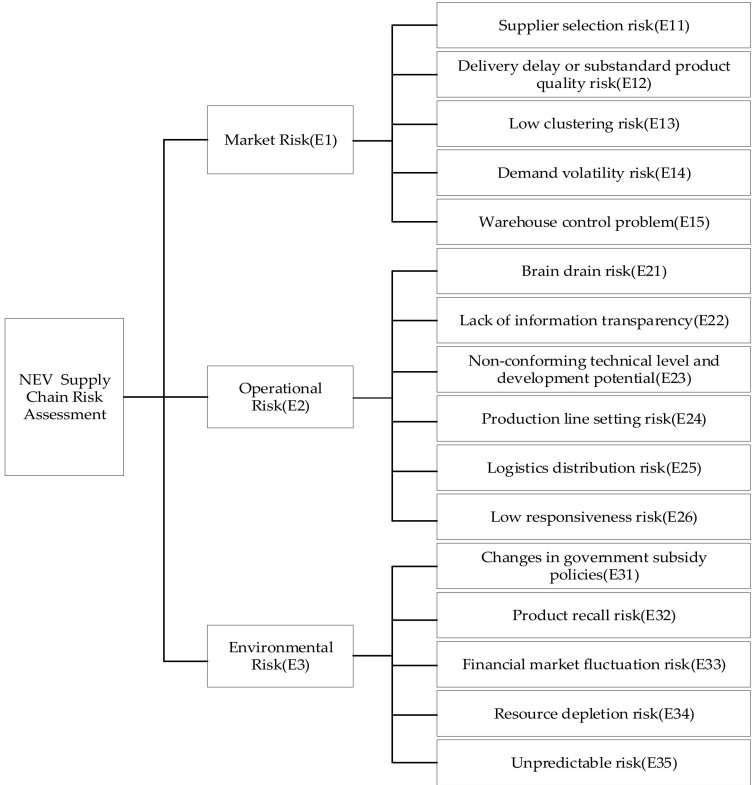

**Figure 1.** New energy vehicle (NEV) supply chain risk assessment criteria system.

### 3.1.3. Low Clustering Risk (E13)

The cluster supply chain is the organic integration of industrial cluster and supply chain management. Dealing with the relationship between the whole vehicle enterprise and the component enterprise, so that win–win cooperation is mutually reinforcing, can promote healthy development of the new energy vehicle industry, such as is the case with Toyota. Low clustering will inhibit product innovation and reduce information sharing, resulting in inefficient supply chain operations. In this paper, the low clustering risk is included in the evaluation system for the first time, to comply with the trend of the times and industry development needs [57].

### 3.1.4. Demand Volatility Risk (E14)

Although a number of leading companies such as Tesla have emerged in the field of new energy vehicles, it is still difficult to accurately predict consumer demand. Abundant studies have shown that consumer demand is not only related to itself, but also government policies and supplier service levels, etc. [19,21]. The uncertainty of consumer demand will bring about a whole string of shock to the supply chain, and it will also impose higher requirements on the flexible production capacity of each node company.

### 3.1.5. Warehouse Control Problem (E15)

Reasonable inventory arrangements have considerable influence on lean production. In order to cope with the waste of materials in the production process, inventory control should be within a logical range, otherwise unnecessary inventory expenditures will be raised, which will aggravate the capital burden of the supply chain [22,28].

### 3.2. Operational Risk (E2)

These criteria refer to the negative damage to the supply chain triggered by the irrationality of control and operation, which hinders its sustainability [22,28,34].

### 3.2.1. Brain Drain Risk (E21)

Talent is the driving force of new energy vehicle production to make considerable strides and the core of modern social competition. Brain drain subjoins corporate training costs, but may also lead to technology leakage and job vacancies. Therefore, it would cause great disruption to the smooth operation of the production line, and undermine the favoring functioning of the supply chain [28,34].

### 3.2.2. Lack of Information Transparency (E22)

Information transparency implies that the information of the suppliers and products are readily available for the end users and other companies [22]. Information sharing of node enterprises is the essential element to realize the integration and unification of the supply chain. Only changing the consciousness of "one-way advantage" among enterprises can actualize the global optimization of the supply chain. A lack of information transparency leads to the inconsistency of decision-makers' actions in all aspects of the supply chain, hinders the information transmission, and depresses the control ability of the supply chain. Then, it further forces members to rely on forecasting and establishing safety stock, which will make the situation worse and augment the vulnerability of the supply chain in the absence of visualization.

### 3.2.3. Non-Conforming Technical Level and Development Potential (E23)

In order to lessen climate change, abate pollution, and effectively exploit natural resources, the government began to vigorously develop new energy vehicles. Compared with traditional vehicles, the technical barriers of NEV are higher, and the benchmarks for parts are more stringent. After electrification, motor, battery, and electronic controlling account for nearly half of the value of the supply chain, and strengthening the technology of the parts industry has become the lynchpin in terms of blazing new trails in the supply chain. Meanwhile, the construction of the Internet of Vehicles and Automatic Operation also puts forward requirements for the technological level and development potential of various enterprises. Technology plays a vital role in the supply chain management of new energy vehicles and is an extremely critical sub-criterion. Technical failures not only increase the rejection rate and give rise to energy dissipation, but may also harm product delivery. The development of new technologies, such as 5th generation mobile networks, challenges the technical development of NEV. Accordingly, the potential for technological innovation is a safeguard to continuously advance the safety level, efficiency, and quality of the supply chain [22,23,28].

### 3.2.4. Production Line Setting Risk (E24)

To ensure the quality of the final product, original equipment manufacturers (OEMs) strictly control all aspects of the production line in the spirit of excellence. In the design of the production line, an excessive design cycle, insufficient capital investment, and lack of talent may affect the manufacturing time and efficiency, and then supply chain risks may be on the cards. In addition, as a result of the large number of assembly lines involved in the production process of NEVs, equipment failure or program errors in any part may lead to production stagnation and output errors, thereby disrupting the supply chain [34].

### 3.2.5. Logistics Distribution Risk (E25)

The logistics distribution risks between diverse nodes mainly predicate the interruption caused by the choice of transportation methods, the arrangement of transportation batches and quantities, third-party logistics service providers, and other unexpected factors [23,34]. An unreasonable design of

the logistics distribution network may initiate delays in delivery, which, in turn, will cause out-of-stock losses, and thus, would bring a poor consumer experience to consumers, adversely exerting a bad influence on car brands.

### 3.2.6. Low Responsiveness Risk (E26)

Fluctuations in demand for new energy vehicles require the supply chain to be able to respond quickly, and all procedures must work together to meet the individual and diverse needs of consumers. The main strength of rapid response is its contingency philosophy, which requires that the parts depending on multiple settings are closely connected to improve the ability to respond to changes in user needs [28,34,57].

### 3.3. Environmental Risk (E3)

Environmental risk refers to the impact of uncertainty in the natural and social environment on the supply chain, such as changes in policies, laws, and emergencies [22,28,30].

### 3.3.1. Changes in Government Subsidy Policies (E31)

In the early stage of the development of NEVs, government subsidies were a significant factor in consumer purchase demand. Changes in subsidy intensity will lead to alterations in the price advantage of new energy vehicles and impact on sales volume [22,58].

### 3.3.2. Product Recall Risk (E32)

This sub-criterion refers to the risk that a sudden increase in the supply chain cost in the process of product recall may cause the supply chain to face the risk of benefit decline, and affect the brand image of automobiles, thus resulting in venture to the terminal sales link of the supply chain [34].

### 3.3.3. Financial Market Fluctuation Risk (E33)

As a result of the large initial investment in the new energy vehicle supply chain, companies will conduct financing in conjunction with specific business models in the operation process [28,34]. However, owing to various uncertain factors that cannot be predicted in advance, such as market price fluctuation and exchange rate change, it will deviate from the expectation and even suffer losses.

### 3.3.4. Resource Depletion Risk (E34)

With the increase in the number of NEVs, the global competition for metal resources is also hotting up. In China, the main power batteries, i.e., lithium and cobalt, are short of resources, and the challenges associated with stable supply of resources and stable price are great [26,27]. At the same time, the problems of power battery recycling and electricity cleaning are increasingly prominent.

### 3.3.5. Unpredictable Risk (E35)

Unpredictable risks mainly refer to force majeure factors, that is, objective conditions that cannot be foreseen, avoided, or overcome by manpower, including certain natural phenomena, such as earthquakes, typhoons, floods, tsunamis, and some social phenomena, such as war, etc. [26].

## 4. Methodology

In order to avoid the subjectivity and limitation of the decision-maker, this paper uses fuzzy analytic hierarchy process (FAHP) to determine the constant weight of risk factors. Then we propose to use the variable weight formula of incentive mechanism to modify the constant weight, which can reflect the balance of each level in the comprehensive evaluation. The combination of the two methods can avoid the abnormal data of critical criterion, which cannot be fully reflected in the evaluation results. Considering the randomness and fuzziness of expert evaluation data, this paper applies cloud

model to NEV supply chain risk evaluation based on the above, which can solve the problem of a too rigid boundary of level interval division. The detailed risk assessment process is shown in Figure 2.

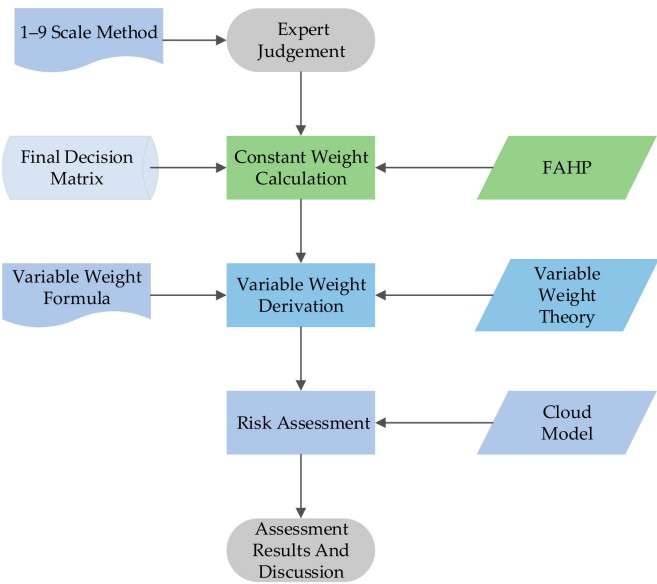

**Figure 2.** Risk assessment process.

### 4.1. Fuzzy Analytic Hierarchy Process (FAHP)

Currently, the methods used to determine the weights of criteria in comprehensive evaluations are divided into subjective weights, based on expert experience, and objective weights, based on objective data. As a result of the fuzziness and non-determinacy of supply chain risk assessment, this paper selects the FAHP as the method of constant weight determination. AHP was first proposed by operations researcher, T.L. Saaty, in the early 1970s [59]. This method analyzes qualitative factors quantitatively, which can predigest involved problems. Due to the ambiguity of human subjective judgment, there are some limitations in the application of AHP. Van Laarhoven, a Dutch scholar, uses triangular fuzzy number (TFN) to extend the traditional AHP to fuzzy AHP, which improves its scientificity and rationality [60,61].

4.1.1. Establishing a Hierarchical Model and Judgement Matrix

Generally, the hierarchy model is split into three units, namely, the target layer, criteria, and sub-criterion, which is the basic framework of the subsequent work. For the purpose of quantifying analysis results, this paper uses the 1–9 scale method shown in Table 1 as the principle for experts to make judgments on factors. The scoring results are converted into triangular fuzzy numbers to establish a fuzzy judgment matrix.

**Table 1.** 1–9 scale of the analytic hierarchy process (AHP).

| Score | Implication |
| --- | --- |
| 1 | The two elements are of equal significance. |
| 3 | The former is slightly more significant than the latter. |
| 5 | The former is more significant than the latter. |
| 7 | The former is intensively more significant than the latter. |
| 9 | The former is extremely more significant than the latter. |
| 2,4,6,8 | Median of the above adjoining judgments. |

Furthermore, the expert scoring results are transformed into triangle fuzzy numbers to set up a fuzzy judgment matrix [62]. Assuming that $\widetilde{t}_{ij} = \left(t_{ij}^L, t_{ij}^M, t_{ij}^U\right)$, $t_{ij}^L \leq t_{ij}^M \leq t_{ij}^U$, where $t_{ij}^L, t_{ij}^M$ and $t_{ij}^U$ indicate the lower limit, mid-value, and top limit, respectively. It can be calculated using Equation (1).

$$\begin{cases} t_{ij}^L = Min\left(t_{ijk}\right) \\ t_{ij}^M = \sqrt[n]{\prod_{k=1}^{n} t_{ijk}} \\ t_{ij}^U = Max\left(t_{ijk}\right) \end{cases}, \tag{1}$$

where $t_{ijk}$ is defined as the judgment score of the k-th expert for the relative significance of elements $i$ and $j$.

Hence, the final fuzzy judgment matrix $\widetilde{T}$ described by TFNs is shown as follows:

$$\widetilde{T} = \left[\widetilde{t}_{ij}\right] = \begin{bmatrix} (1,1,1) & \widetilde{t}_{12} & \cdots & \widetilde{t}_{1n} \\ \widetilde{t}_{21} & (1,1,1) & \cdots & \widetilde{t}_{2n} \\ \vdots & \vdots & \vdots & \vdots \\ \widetilde{t}_{n1} & \widetilde{t}_{n2} & \cdots & (1,1,1) \end{bmatrix} \tag{2}$$

Defuzzification is fully critical with regard to the convenient application in non-fuzzy environments. After the comparative analysis and expert consultation, this paper employs Equations (3)–(6) to remove fuzziness of the matrix $\widetilde{T}$ [63].

$$\left(t_{ij}^\lambda\right)^\delta = \left[\delta \cdot \left(t_{ij}^L\right)^\lambda + (1-\delta) \cdot \left(t_{ij}^U\right)^\lambda\right] \tag{3}$$

$$\left(t_{ij}^\lambda\right)^\delta = 1 / \left(t_{ji}^\lambda\right)^\delta \tag{4}$$

$$\left(t_{ij}^L\right)^\lambda = \left(t_{ij}^M - t_{ij}^L\right) \cdot \lambda + t_{ij}^L \tag{5}$$

$$\left(t_{ij}^U\right)^\lambda = t_{ij}^U - \left(t_{ij}^U - t_{ij}^M\right) \cdot \lambda \tag{6}$$

$\lambda$ reveals the preference coefficient of the estimator, and the smaller the value, the greater the non-determinacy of the judgment. On the other hand, $\delta$ expresses the risk tolerance of the evaluator, and the higher the value, the more pessimistic the evaluator. Both are values between 0 and 1. In this paper, their values are both 0.5. The final decision matrix is as follows:

$$\left(T^\lambda\right)^\delta = \left[\left(t_{ij}^\lambda\right)^\delta\right] = \begin{bmatrix} 1 & \left(t_{12}^\lambda\right)^\delta & \cdots & \left(t_{1n}^\lambda\right)^\delta \\ \left(t_{21}^\lambda\right)^\delta & 1 & \cdots & \left(t_{2n}^\lambda\right)^\delta \\ \vdots & \vdots & \vdots & \vdots \\ \left(t_{n1}^\lambda\right)^\delta & \left(t_{n2}^\lambda\right)^\delta & \cdots & 1 \end{bmatrix} \tag{7}$$

### 4.1.2. Hierarchical Single Ordering and Consistency Test

On account of the various criteria involved in this paper, this paper will use MATLAB software to compute the maximum eigenvalue $\lambda_{\max}$ and eigenvector $\omega$ of the matrix, and then normalize them to obtain the weight order of each single level of internal elements. In addition, the consistency index *CR*

will be calculated from the following equation. If its value is less than 0.1, it shows that it satisfies the consistency test, otherwise there is a need for revision.

$$CR = {}^{CI}\!/_{RI} \tag{8}$$

$$CI = (\lambda_{\max} - m)/(m-1) \tag{9}$$

At length, the final constant weight is derived through the total sorting process.

### 4.2. Variable Weight Theory

Although constant weight can well reflect the relative importance of each sub-criterion, its weight value will not change due to the different states of affairs. In the risk assessment of the NEV supply chain, if the value of a criterion deviates from the normal value, it will bring greater risk to the supply chain, but in the constant weight paradigm, it may not reflect the real state of the supply chain because of its small weight. The essence of variable weight theory is to introduce a state variable weight vector on the basis of a constant weight vector, so as to ensure that the weight value can be changed according to the state value of factors or the diversification of specific circumstances.

According to the monotonicity of variable weight, this falls into two types: the penalty variable weight and the incentive variable weight. For the supply chain risk assessment, the larger the status value of each element, the more serious the risk will be, and the worse the risk assessment result will be. Therefore, this paper applies the variable weight formula of incentive mechanism to modify the constant weight [64–66].

$$\omega_j^v(x) = \frac{\omega_j x_j^\alpha}{\sum\limits_{j=1}^{m} \omega_j x_j^\alpha} \tag{10}$$

Among them, $\alpha$ is the equilibrium coefficient, which satisfies the condition $0 \leq \alpha \leq 1$, indicating the proportionate degree among the criteria. $\omega_j^v, \omega_j$ and $x_j$ indicate variable weight, constant weight, and normalized value of the index $j(j = 1 \cdots m)$, respectively.

### 4.3. Cloud Model

The cloud model is a kind of uncertainty model which can realize the transformation between a qualitative concept and a quantitative value, and combines the fuzziness and randomness of things. The normal cloud model is one of the commonly used cloud models, which is expressed by three numerical characteristics: expectation $E_x$, entropy $E_n$ and super entropy $H_e$. The expectation is the most representative value of a qualitative concept, while the entropy reflects the fuzziness of the attribute concept, and super entropy describes the dispersion degree of the cloud droplet distribution, that is, the randomness [67,68].

The first step in using cloud model for assessment is to establish the evaluation cloud model. The evaluation value domain of the criteria can be split into multiple areas in accordance with the evaluation level number, and then the digital eigenvalue of each level cloud model can be calculated by the positive cloud generator [69,70]. See Equation (11) for details.

$$\begin{cases} E_{xr} = (x_r^{\max} + x_r^{\min})/2 \\ E_{nr} = (x_r^{\max} - x_r^{\min})/6 \quad , \\ \quad H_{er} = z \end{cases} \tag{11}$$

Among them, $r$ represents the divided level interval, and $z$ can be adjusted according to the actual situation of corresponding criterion.

Next, the basic cloud model $M_j\left(E_{xj}, E_{nj}, H_{ej}\right)$ is established according to the evaluation scores of the following equation [71].

$$
\begin{cases}
E_{xj} = \frac{1}{n}\sum_{i=1}^{n} x_{ij} \\
E_{nj} = \sqrt{\frac{\pi}{2}} \times \frac{1}{n}\sum_{i=1}^{n} \left| x_{ij} - E_{xj} \right| \\
H_{ej} = \sqrt{\left| S_j^2 - E_{nj}^2 \right|} \\
S_j^2 = \frac{1}{n-1}\sum_{i=1}^{n} \left( x_{ij} - \overline{X}_j \right)^2
\end{cases}
\tag{12}
$$

Finally, integrated with the variable weight obtained above, the comprehensive cloud $M(E_x, E_n, H_e)$ can be calculated [72].

$$
\begin{cases}
E_x = \dfrac{\sum_{j=1}^{m} E_{xj} E_{nj} \omega_j^v}{\sum_{j=1}^{m} E_{nj} \omega_j^v} \\
E_n = \sum_{j=1}^{m} E_{nj} \omega_j^v \\
H_e = \dfrac{\sum_{j=1}^{m} H_{ej} E_{nj} \omega_j^v}{\sum_{j=1}^{m} E_{nj} \omega_j^v}
\end{cases}
\tag{13}
$$

By calculating the similarity degree between the grade cloud model and the comprehensive cloud, the comprehensive cloud model can be distinguished, and then the maximum similarity principle and the cloud map realized by MATLAB. These can then be used for evaluation [73].

## 5. Case Study

This paper invited experts to conduct a risk assessment on China's NEV supply chain. Experts from universities, the Future Science City of Changping District in Beijing, China, and upstream and downstream companies from the NEV supply chain were selected, and invitations for a questionnaire survey were sent to them. These experts either had conducted research related to new energy vehicles, especially supply chain management, or had a rich amount of experience working in relevant enterprises. All of them had at least five years of project research experience, and most of them had doctoral degrees or advanced professional titles. After obtaining their permission, this paper collected experts' opinions through online questionnaires. Experts were required to evaluate the relative importance of each criterion according to the 1–9 scale method; on the other hand, they also needed to estimate the risk status of China's NEV supply chain. It is stated in advance that this paper divides the risk assessment level into 5 grades using the score value, that is, the assessment set is V = {V1, V2, V3, V4, V5} = {Very Low; Low; General; High; Very High}.

*5.1. Determination of Weight*

5.1.1. Constant Weight Recalculation

Experts judged the relative importance of each criterion according to the 1–9 scale method and the evaluation data obtained were processed according to Equations (1)–(6). Taking the weight acquisition of the criteria layer to the target layer as an example, the fuzzy judgment matrix was obtained through data handling, and then the maximum eigenvalue and eigenvector of the fuzzy judgment matrix were

calculated through defuzzification and using the MATLAB software. The fuzzy judgment matrix and the de-fuzzy judgment matrix are shown in Equations (14) and (15).

$$\widetilde{E}_0 = \begin{pmatrix} & E1 & E2 & E3 \\ E1 & (1,1,1) & (0.33,0.47,0.5) & (1,1.21,2) \\ E2 & (2,2.15,3) & (1,1,1) & (1,2.35,3) \\ E3 & (0.5,0.89,1) & (0.33,0.76,1) & (1,1,1) \end{pmatrix} \tag{14}$$

$$E_0 = \begin{pmatrix} & E1 & E2 & E3 \\ E1 & 1.00 & 0.44 & 1.36 \\ E2 & 2.26 & 1.00 & 2.18 \\ E3 & 0.74 & 0.46 & 1.00 \end{pmatrix} \tag{15}$$

The maximum eigenvalue counted by MATLAB was $\lambda^0_{max} = 3.014$. Furthermore, $CR = 0.012 < 0.1$, which proves that the matrix passes the consistency test. According to the corresponding eigenvector, the weight vector is $\omega^0 = (0.26, 0.52, 0.22)$. The constant weights of the other criteria at all levels is also based on the above process, and the specific results are shown in Equations (16)–(21).

$$\widetilde{E}_1 = \begin{pmatrix} & E11 & E12 & E13 & E14 & E15 \\ E11 & (1,1,1) & (2,2.49,3) & (0.33,0.43,0.5) & (1,1.12,2) & (2,3.01,4) \\ E12 & (0.33,0.42,0.5) & (1,1,1) & (0.2,0.21,0.25) & (0.33,0.39,0.5) & (1,1.19,2) \\ E13 & (2,2.49,3) & (4,4.76,5) & (1,1,1) & (2,2.91,3) & (3,3.69,4) \\ E14 & (0.5,0.88,1) & (2,2.51,3) & (0.33,0.4,0.5) & (1,1,1) & (1,1.91,3) \\ E15 & (0.25,0.37,0.5) & (0.5,0.79,1) & (0.25,0.29,0.33) & (0.33,0.61,1) & (1,1,1) \end{pmatrix} \tag{16}$$

$$E_1 = \begin{pmatrix} & E11 & E12 & E13 & E14 & E15 \\ E11 & 1.00 & 2.50 & 0.42 & 1.31 & 3.01 \\ E12 & 0.40 & 1.00 & 0.22 & 0.40 & 1.35 \\ E13 & 2.37 & 4.60 & 1.00 & 2.71 & 3.60 \\ E14 & 0.76 & 2.48 & 0.37 & 1.00 & 1.96 \\ E15 & 0.33 & 0.74 & 0.51 & 0.51 & 1.00 \end{pmatrix} \tag{17}$$

$$\widetilde{E}_2 = \begin{pmatrix} & E21 & E22 & E23 & E24 & E25 & E26 \\ E21 & 1,1,1 & 0.17,0.21,0.25 & 0.14,0.25,0.33 & 0.33,0.45,0.5 & 1,1.21,2 & 0.14,0.17,0.2 \\ E22 & 4,4.76,6 & 1,1,1 & 1,1.21,2 & 2,2.79,3 & 3,4.13,5 & 1,1.13,2 \\ E23 & 3,5.17,7 & 0.5,0.89,1 & 1,1,1 & 3,3.31,4 & 4,4.71,5 & 1,1.33,3 \\ E24 & 2,2.71,3 & 0.33,0.41,0.5 & 0.25,0.29,0.33 & 1,1,1 & 2,2.91,4 & 0.25,0.41,0.5 \\ E25 & 0.5,0.96,1 & 0.2,0.27,0.33 & 0.2,0.22,0.25 & 0.25,0.34,0.5 & 1,1,1 & 0.17,0.19,0.2 \\ E26 & 5,6.32,7 & 0.5,0.91,1 & 0.33,0.86,1 & 2,3.33,4 & 5,5.71,6 & 1,1,1 \end{pmatrix} \tag{18}$$

$$E_2 = \begin{pmatrix} & E21 & E22 & E23 & E24 & E25 & E26 \\ E21 & 1.00 & 0.21 & 0.24 & 0.43 & 1.36 & 0.17 \\ E22 & 4.76 & 1.00 & 1.36 & 2.65 & 4.07 & 1.32 \\ E23 & 4.12 & 0.74 & 1.00 & 3.41 & 4.61 & 1.67 \\ E24 & 2.31 & 0.38 & 0.29 & 1.00 & 2.96 & 0.39 \\ E25 & 0.74 & 0.25 & 0.22 & 0.34 & 1.00 & 0.19 \\ E26 & 5.88 & 0.76 & 0.60 & 2.55 & 5.33 & 1.00 \end{pmatrix} \tag{19}$$

$$\widetilde{E}_3 = \begin{array}{c} \\ E31 \\ E32 \\ E33 \\ E34 \\ E35 \end{array} \left( \begin{array}{ccccc} E31 & E32 & E33 & E34 & E35 \\ (1,1,1) & (4,4.21,5) & (2,2.43,3) & (3,3.85,5) & (4,5.76,7) \\ (0.2,0.23,0.25) & (1,1,1) & (0.33,0.37,0.5) & (1,1.56,3) & (1,1.59,2) \\ (0.33,0.41,0.5) & (2,2.71,3) & (1,1,1) & (2,2.73,4) & (2,2.22,3) \\ (0.2,0.27,0.33) & (0.33,0.71,1) & (0.25,0.44,0.5) & (1,1,1) & (1,1.71,2) \\ (0.14,0.19,0.25) & (0.5,0.78,1) & (0.33,0.39,0.5) & (0.5,0.87,1) & (1,1,1) \end{array} \right) \tag{20}$$

$$E_3 = \begin{array}{c} \\ E31 \\ E32 \\ E33 \\ E34 \\ E35 \end{array} \left( \begin{array}{ccccc} E31 & E32 & E33 & E34 & E35 \\ 1.00 & 4.36 & 2.47 & 3.93 & 5.63 \\ 0.23 & 1.00 & 0.39 & 1.78 & 1.55 \\ 0.41 & 2.55 & 1.00 & 2.87 & 2.36 \\ 0.25 & 0.56 & 0.35 & 1.00 & 1.61 \\ 0.18 & 0.65 & 0.42 & 0.62 & 1.00 \end{array} \right) \tag{21}$$

The maximum eigenvalues calculated from each matrix are 5.278, 6.140, and 5.092. The values of CR are 0.062, 0.023, and 0.021, respectively, showing that they all passed the consistency test. The final result of constant weight is displayed in Table 2.

**Table 2.** Constant weight.

| Criterion | Weight Relative to Target Layer | Sub-Criterion | Weight Relative to Criterion Layer | Constant Weight |
|---|---|---|---|---|
| E1 | 0.260 | E11 | 0.219 | 0.057 |
|  |  | E12 | 0.090 | 0.023 |
|  |  | E13 | 0.414 | 0.108 |
|  |  | E14 | 0.174 | 0.045 |
|  |  | E15 | 0.103 | 0.027 |
| E2 | 0.525 | E21 | 0.054 | 0.028 |
|  |  | E22 | 0.277 | 0.146 |
|  |  | E23 | 0.275 | 0.144 |
|  |  | E24 | 0.106 | 0.056 |
|  |  | E25 | 0.049 | 0.026 |
|  |  | E26 | 0.238 | 0.125 |
| E3 | 0.215 | E31 | 0.470 | 0.101 |
|  |  | E32 | 0.120 | 0.026 |
|  |  | E33 | 0.235 | 0.051 |
|  |  | E34 | 0.096 | 0.021 |
|  |  | E35 | 0.078 | 0.017 |

### 5.1.2. Derivation of Variable Weight

According to the experts' risk evaluation scores, the state vector was constructed, and the constant weight was modified in accordance with Equation (10). The value of the equalization coefficient was set to 0.5. The variable weight of each sub-criterion was obtained as follows:

$$\omega_1^v = (0.055, 0.022, 0.108, 0.045, 0.026) \tag{22}$$

$$\omega_2^v = (0.027, 0.148, 0.149, 0.055, 0.024, 0.128) \tag{23}$$

$$\omega_3^v = (0.105, 0.024, 0.051, 0.019, 0.013) \tag{24}$$

### 5.2. Cloud Model Construction

On the basis of the structured criteria system, a set of comments was established as V = {V1, V2, V3, V4, V5} = {Very Low; Low; General; High; Very High}, considering the relative gap between the existing supply chain and the transformation goal. This paper first conducts the cloud models of all

levels using Equation (11), and then the corresponding eigenvalues and evaluation cloud charts are shown in Table 3 and Figure 3.

**Table 3.** Eigenvalues of evaluating cloud model for NEV supply chain risk.

| Grade | Cloud Model |
|---|---|
| Very Low (V1) | (15, 5, 0.5) |
| Low (V2) | (42.5, 4.2, 0.5) |
| General (V3) | (65, 3.3, 0.5) |
| High (V4) | (82.5, 2.5, 0.5) |
| Very High (V5) | (95, 1.7, 0.5) |

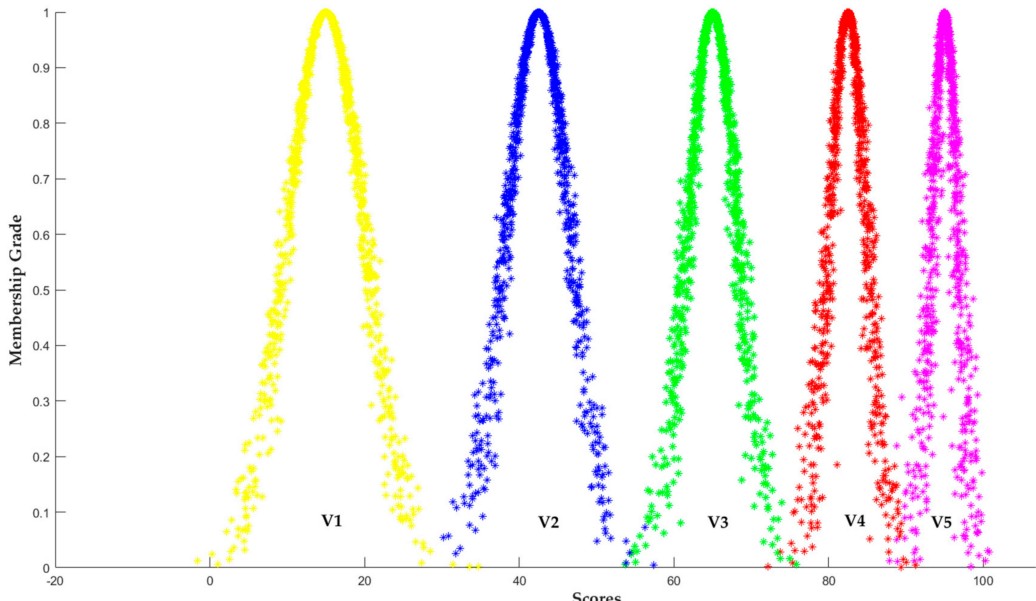

**Figure 3.** Cloud chart of evaluating grades.

Furthermore, the comprehensive cloud under variable weight was calculated using the comprehensive cloud algorithm and expert evaluation results; that is, $M^v = (84.964, 2.291, 0.595)$. According to the cloud similarity algorithm, we calculated the similarity between the comprehensive cloud and each evaluation level cloud under the variable weight as shown in Table 4; Figure 4 is the comprehensive cloud map generated using the cloud generator.

**Table 4.** Similarity evaluation results.

| Grades | Very Low (V1) | Low (V2) | General (V3) | High (V4) | Very High (V5) | Results |
|---|---|---|---|---|---|---|
| Cloud model | (15, 5, 0.5) | (42.5, 4.2, 0.5) | (65, 3.3, 0.5) | (82.5, 2.5, 0.5) | (95, 1.7, 0.5) | |
| Comprehensive cloud | | | (84.964, 2.291, 0.595) | | | |
| Euclidean distance | 70.017 | 42.507 | 19.990 | 2.475 | 94.939 | High (V4) |
| Membership degree of fuzzy synthesis evaluation | 0 | 0 | 0.034 | 0.729 | 0.311 | High (V4) |

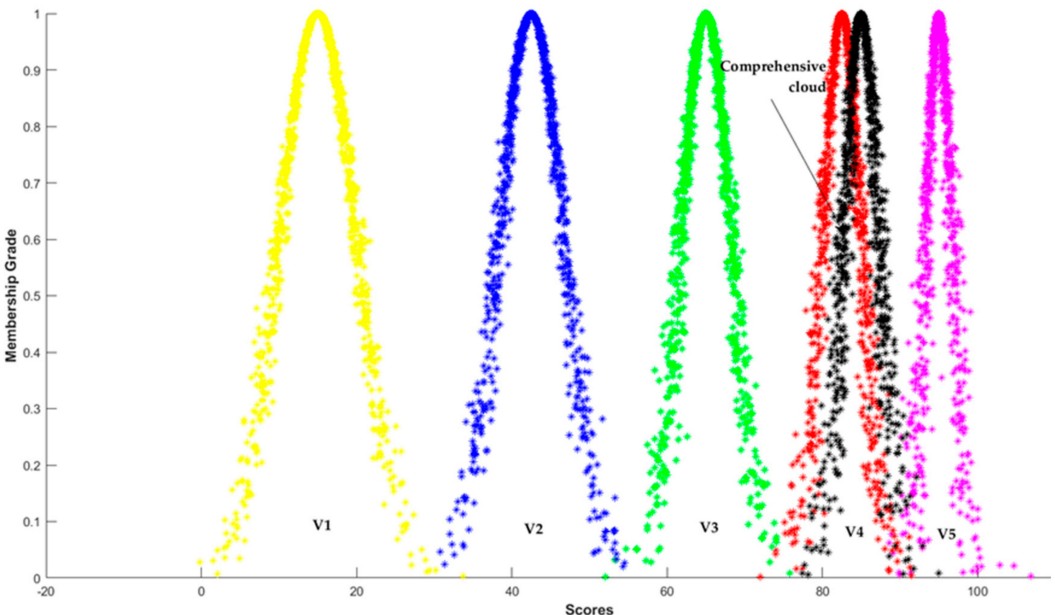

**Figure 4.** Comprehensive cloud with variable weight.

*5.3. Discussion*

5.3.1. Discussion on the Assessment Results

From the results of cloud model with variable weight, we can draw the conclusion that the risk of China's NEV supply chain is at a high level. In the stage of the transformation of the automobile industry to electrification, there is a growing call to attach importance to supply chain management, and the results demonstrate that the current risks are mainly rooted in the market and operation phases. This is consistent with the conclusions from the literature [26]. In order to achieve the goal of environmental protection, energy conservation, and emission reduction, new energy vehicles have emerged as the times require. However, in the process of reconstruction, the change in the market environment makes the market demand more difficult to predict, which brings challenges to the supply chain. This step from the non-growing industrial cluster and long cycle of the supply chain shows that the operation efficiency of supply chain is relatively poor, and it seriously lacks stability.

For market risks, the weight of the low clustering is the highest, closely followed by the supplier selection risk, and demand volatility risk. The cluster development of the supply chain is the key to raising efficiency and a major requisite to achieving lean management, so it occupies an important share. The increasingly fierce market competition requires a careful selection of suppliers to satisfy the requirements of time and quality. Meanwhile, the changing market has become a hard nut to crack for demand forecasting.

In terms of the operational risk, although the production line, logistics distribution, and talents have a vital function in supply chain management, the technical level, information transparency, and response ability have a far-reaching impact on the supply chain. The optimization of NEV motor configuration and charging pile design throws down the gauntlet to the technical ability of each node enterprise in the supply chain, which is not only the problem of talent management, but also needs innovation and change. Low information transparency brings randomness and uncertainty to all links, while responsiveness is the necessity for the supply chain to reply to the changes in the market in a quick and flexible way.

Regarding environmental risks, changes in government subsidy policies holds the maximum weight. The government subsidy policy plays a guiding role in the new energy vehicle industry. The decline or even withdrawal of the government subsidy policy in China has brought great volatility and uncertainty to the new energy vehicle market.

### 5.3.2. Comparative Analysis

This paper calculates the comprehensive cloud based on constant weight, and the result is $M^c = (84.680, 2.285, 0.5999)$. Compared with the comprehensive cloud with variable weight, the modified variable weight causes the risk value to increase slightly. This is due to the fact that this paper adopts the conservative incentive variable weight method, which can overcome the non-transferable attribute value by reflecting the psychological characteristics of decision-makers [66]. Variable weight well reflects the impact of the numerical changes of key criterion on the risk status of supply chain.

Under the variable weight model, the fuzzy synthesis evaluation method introduced in [26] was used to assess China's supply chain risk to verify the effectiveness of the method. Table 4 reveals the evaluation results of the fuzzy synthesis evaluation method, showing no difference with that of the cloud model. Obviously, compared with the fuzzy synthesis evaluation method, the cloud model can reflect the randomness of things and give expression to the essence of fuzziness. In the parameters of the comprehensive cloud, the super entropy is 0.595. This means that the current risk state is relatively stable, indicating that China's NEV supply chain will continue to remain in the high-risk status for a period of time in the future. This is mainly because China's electric and intelligent transformation is still in progress, and there is a long time to build a mature new energy vehicle supply chain [5]. However, these trends cannot be seen from the results of the fuzzy synthesis evaluation. Therefore, the cloud model can provide more information for risk assessment.

### 5.3.3. Sensitivity Analysis

In the process of risk assessment of the NEV supply chain, the weight of evaluation criteria is one of the major contributors leading to the uncertainty of the results. Although the weight of each evaluation criterion calculated based on FAHP method was adjusted by the theory of variable weight, its result will be affected to a certain extent by the preference, knowledge, and experience of decision-makers. Therefore, this section verifies the evaluation results through sensitivity analysis.

This paper sets up the following four scenarios. In addition, the weight of other criteria should be adjusted proportionally to ensure that the sum of weights is 1.

Scenario 1. Decreasing the weight of one criterion by 10%;
Scenario 2. Decreasing the weight of one criterion by 20%;
Scenario 3. Increasing the weight of one criterion by 10%;
Scenario 4. Increasing the weight of one criterion by 10%.

The final results are shown in Table 5. The results of Euclidean distance calculation between each comprehensive cloud and evaluation level cloud show that they are all close to V4, which is a high level. It can be seen that, with the change of the weight of each criterion, the evaluation result has a certain stability, revealing that the method used in this paper is relatively robust.

**Table 5.** Sensitivity analysis results.

| Comprehensive Cloud | Scenario 1 | Scenario 2 | Scenario 3 | Scenario 4 |
|---|---|---|---|---|
| E11 | (84.991, 2.295, 0.595) | (85.018, 2.299, 0.596) | (84.937, 2.287, 0.594) | (84.910, 2.282, 0.593) |
| E12 | (84.981, 2.292, 0.595) | (84.999, 2.292, 0.595) | (84.947, 2.290, 0.594) | (84.930, 2.289, 0.594) |
| E13 | (84.972, 2.298, 0.598) | (84.981, 2.305, 0.601) | (84.956, 2.284, 0.591) | (84.947, 2.277, 0.588) |
| E14 | (84.961, 2.290, 0.593) | (84.958, 2.290, 0.591) | (84.967, 2.291, 0.596) | (84.970, 2.292, 0.598) |
| E15 | (84.978, 2.292, 0.594) | (84.993, 2.293, 0.594) | (84.950, 2.290, 0.595) | (84.935, 2.289, 0.595) |
| E21 | (84.976, 2.291, 0.593) | (84.988, 2.291, 0.592) | (84.952, 2.291, 0.596) | (84.940, 2.291, 0.597) |
| E22 | (84.939, 2.303, 0.598) | (84.915, 2.316, 0.601) | (84.989, 2.278, 0.591) | (85.015, 2.266, 0.588) |
| E23 | (84.832, 2.277, 0.599) | (84.699, 2.264, 0.603) | (85.094, 2.304, 0.590) | (85.223, 2.318, 0.586) |
| E24 | (85.007, 2.285, 0.586) | (85.051, 2.279, 0.578) | (84.921, 2.297, 0.602) | (84.879, 2.303, 0.610) |
| E25 | (84.994, 2.291, 0.594) | (85.023, 2.291, 0.593) | (84.935, 2.290, 0.595) | (84.905, 2.290, 0.596) |
| E26 | (84.915, 2.288, 0.593) | (84.867, 2.284, 0.591) | (85.013, 2.294, 0.596) | (85.061, 2.297, 0.598) |
| E31 | (84.885, 2.287, 0.599) | (84.806, 2.284, 0.603) | (85.043, 2.294, 0.590) | (85.121, 2.297, 0.586) |
| E32 | (85.003, 2.290, 0.595) | (85.042, 2.290, 0.596) | (84.925, 2.291, 0.594) | (84.886, 2.291, 0.593) |
| E33 | (84.960, 2.291, 0.594) | (84.956, 2.292, 0.594) | (84.968, 2.290, 0.595) | (84.972, 2.289, 0.595) |
| E34 | (85.000, 2.290, 0.593) | (85.036, 2.289, 0.591) | (84.928, 2.291, 0.596) | (84.892, 2.292, 0.598) |
| E35 | (84.998, 2.291, 0.595) | (85.032, 2.292, 0.595) | (84.930, 2.290, 0.594) | (84.896, 2.289, 0.594) |

### 5.3.4. External Validity Analysis

In order to further verify the external effectiveness of the model, the supply chain of American new energy vehicles was selected for analysis in this paper. The results are shown in Figure 5. According to Figure 5, it is obvious that the supply chain risk of NEV in the U.S. is lower than in China, but it is also at a high level. The primary cause for this phenomenon may be that it is currently in the exploratory stage of electrification. The new power of automobile building represented by Tesla in the U.S. has become the pioneer of the NEV market by taking advantage of its first mover advantage and Internet gene. Compared with the current situation of NEVs in China, Tesla's technological superiority occupies the main position, with high technical barriers and product leadership, such as energy density and charging speed [74,75]. However, it is worth noting that the super entropy of the comprehensive cloud of the NEV supply chain in the U.S. is 1.194, which is much larger than that of China. This is mainly due to the fact that the industrial chain of China's NEV is more complete from a global perspective, and the pattern of all links in the whole supply chain is becoming more and more clear. Conversely, the shortage of raw materials and capacity of lithium batteries in the U.S. has led to its long-term dependence on mineral imports, which is a passive situation in terms of global competition. China produces nearly two-thirds of the world's lithium batteries, compared with just 5% in the United States, according to Benchmark Minerals Intelligence, which tracks lithium and other commodity prices [76]. This makes it urgent for the United States to improve its lithium battery production capacity and conduct research on new battery technologies to break the supply bottleneck. Therefore, the value of Chinese manufacturing assets starts to stand out and its future growth is highly certain. By contrast, although the U.S. government takes various measures to strengthen the self-sufficiency of the NEV supply chain, there is still a great amount of uncertainty in the current trend of global electrification.

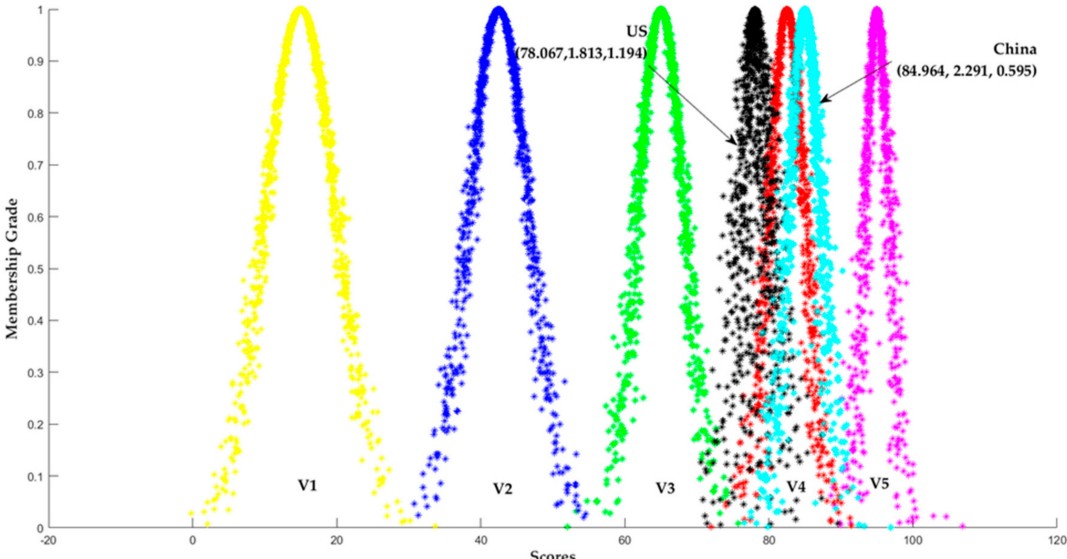

**Figure 5.** External validity analysis results.

## 6. Conclusions

In order to protect the environment and reduce energy consumption, new energy vehicles have begun to be vigorously promoted in various countries. Making innovations in the NEV supply chain has become a very important issue, and its risk assessment is absolutely essential. On the basis of the above analysis, the following conclusions can be reached. Firstly, this paper establishes a risk assessment criteria system of the new energy vehicle supply chain, including market risks, operational risks, and environmental risks. Secondly, the constant weight conducted by FAHP was modified by the state variable, and then the variable weight of each criterion was obtained. It is apparent that the low clustering, lack of information transparency, non-conforming technical level and development

potential, low responsiveness, and changes in government subsidy policies account for relatively large weights in the risk assessment system. Thirdly, the risk assessment of China's NEV supply chain was carried out by building a cloud model based on variable weight. The results show that the risk of China's NEV supply chain is at a high level. Finally, the comparative analysis and external validity analysis demonstrate the validity of the method and reliability of the conclusion.

On the basis of the above analysis results, this paper proposes the following suggestions for the construction of the new energy vehicle supply chain in China. First, the supply chain of new energy vehicles should grow in a cluster form. Only by improving the cluster degree can the efficiency be increased, thereafter, lean management can be realized. Next, we propose establishing Internet cooperation platforms for the supply chain of new energy vehicles to improve the degree of information sharing, so that all links can achieve synergy and integration. Finally, quality assurance should cover all aspects from beginning to end: from the selection of suppliers to the final sales of goods, strict processes and systems should be established to continuously achieve closed-loop quality improvement.

The results of this paper can not only deliver some findings for the development of China's NEV supply chain, but also provide theoretical contributions for the literature. Compared with the previous research, the combination of cloud model and variable weight theory in this paper weakens the randomness and fuzziness of expert decision-making and breaks the absolute limit of level interval division; the existence of super entropy shows that the risk state of China's NEV supply chain is relatively stable at present, and there is a long process to realize electrification in China. Furthermore, low clustering risk and low responsiveness risk newly included in this paper account for a high proportion in the assessment criteria system, and to some extent, the expansion of risk factors regulates the risk state value of China's NEV supply chain.

Despite this paper making certain contributions to the risk assessment of NEV supply chain, there are still some areas that need to be improved. For example, in the process of risk assessment, more attention should be paid to the internal relationship between criteria and their linkage effectiveness. Furthermore, with the gradual deepening of demand-side management, the supply chain of NEV will present more complexity, and its risk management should also be tried in advance and afterwards. Finally, the quantification of the probability and impact of risk events has not been addressed, especially the chain reaction it causes.

**Author Contributions:** Conceptualization, Q.Y.; Formal analysis, M.Z.; Methodology, M.Z.; Writing—original draft, M.Z.; Writing—review and editing, M.Z., W.L. and G.Q. All authors have read and agreed to the published version of the manuscript.

**Funding:** This research was funded by National Social Science Foundation of China, grant number 19ZDA081 and the 2018 Key Projects of Philosophy and Social Sciences Research, Ministry of Education, China, grant number 18JZD032.

**Acknowledgments:** We appreciate the reviewers and editors for their constructive comments.

**Conflicts of Interest:** The authors declare no conflict of interest.

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
