# Peer review of "Risk Assessment of New Energy Vehicle Supply Chain Based on Variable Weight Theory and Cloud Model: A Case Study in China"

_sustainability, doi:10.3390/su12083150_

Round 1

Reviewer 1 Report

A native proofreader is required to check the English.

The authors should explain if cloud model is a technique of sensivity analysis?

Reviewer 2 Report

Thank you very much for the invitation to review this article dealing with risks analysis in new energy vehicles supply chain. Even though the article is interesting in its current format, critical issues should be improved for possible publication and for a better understanding by the readers. My comments and suggestions are provided below:

Focusing my attention on the Abstract. “The results show that China's new energy vehicle supply chain is at a high level.”. What does the following sentence mean? This sentence should be reworded to gain better understanding about the results.

Pages 70-74. “On the other hand, for the methods of risk assessment, scholars generally adopt analytic hierarchy process (AHP [27], TOPSIS[28] and fuzzy comprehensive evaluation[25, 29]. However, most of studies only fasten on the single risk factor, the comprehensive assessment exclusively for China’s NEV supply chain risk in the transition period is rarely involved.”

Methods such as AHP and TOPSIS compute multiple criteria in risk assessment. The authors should briefly discuss earlier articles which only consider single risk factor in the research. Moreover, in the introduction section, the authors should touch upon other methods which consider multiple criteria in supply chain risk analysis.

In my opinion, a great weakness of the research lies in treating supply chain risk analysis as a novel theme, when it has a history of more than 2 decades. In order to overcome this issue, the authors should create a section 2: literature review. Around two subsections, the authors should discuss advances in new energy vehicle supply chain (subsection1) and advances in the analysis of supply chain risk. Please finds some articles below:

https://doi.org/10.3390/su12010154

https://doi.org/10.1016/j.ijpe.2014.11.013

https://doi.org/10.1080/13669877.2012.666757

https://doi.org/10.1016/j.compind.2005.11.001

https://doi.org/10.1016/j.resconrec.2015.01.001

https://doi.org/10.1007/s10479-013-1386-4

FAHP represents greatly consolidated technique. Is really needed coupled FAHP with variable weight theory and a cloud model? This has not been properly explained. Moreover, a figure would help to gain better understanding about how the proposed method is applied. Please include it in Section 3: methodology.

Pages 92-93 and Pages 305-307. It has been not provided any information about expert profile One cannot exclude the possibility of group-think. It is common in professional communities that subgroups share convictions which prove to be wrong when analyzed carefully. Isn’t it one of the duties of academics to critically assess such convictions? Please, provide some information on the type of employer, the country/region where they work and the positions of the experts.

Section 4. Case study. In my opinion, important information about case study is missing. This section does not provide the reader with a clear understanding of structure of studied supply chain, main actors and flows…etc. I am very skeptical about the generalizability their result with only one case study and the contribution it makes. In this way, expert selection must have been carried out carefully. Please explain why the participants were considered experts.

CR values are extremely low. Would you please explain how expert opinions were aggregated?

The analysis and discussion are mere descriptions of the results without the use of theories to highlight how their results are supposed to contribute to the literature.

Limitations of the research should be written as at least a separate paragraph in conclusion section.

Reviewer 3 Report

The content of paper meaningless.

Reviewer 4 Report

The paper proposes a modified methodology to evaluate risk assessment in energy vehicle supply chain. The new method modifies the constant weights of the ‘fuzzy analytic hierarchy process’ by considering weight theory and introduces numerical characteristics (expectation, entropy and super entropy) from a cloud model to quantify risks on the basis of qualitative measures. 

The paper presents an application to study the risks related to the transformation of automobile industry towards electrification in China. For this purpose, three main sources of risk are identified (market, operational and environmental) and classified by 20 experts according to 16-sub-criteria. With such data, the constant and variable weight models are compared finding that the latter induce higher risks. Furthermore, the new model seems not to be significantly different from the fuzzy synthesis evaluation model by Wu et al. (2019). Consequently, interesting conclusions about the current risks from the electrification of automobile industry in China can be extracted.   

The paper deals with hot topic (risk assessment of electricity vehicle supply) but with a rather simplistic methodology. Although in my opinion, the contribution is not outstanding, I also believe that it might be interesting for potential readers of Sustainability, given the importance of the understanding of risks in automobile sector in the Global economy. The paper is well-written and easy-reading, which is another advantage for the readers. However, I enclose a few comments that can be considered in the revision.

Comments

  • Results rely on the opinion of a panel of 20 experts. This seems to be a quite low sample. Even more, few details are given about how the sample and the reported information were obtained. Where all the experts Chinese? As a robustness check results should be validated by increasing the sample (maybe with a panel from other country, as mentioned in point 2).
  • In order to have external validity of the results, I recommend to include an application to the analysis of the risks of the electrification of the automobile sector in other economies (e.g. US or EU) where the process maybe at a different stage than in China. These comparisons would add a significant value to the paper.
  • The title of the paper is too long and unclear. Definitely, it has to be changed.
  • The cites with the authors’ names should be accompanied with a number as well. Even more, the references at the end of the paper appear with “et al.” and should be appropriately cited (see e.g. the reference below).

References

Wu, Y., Jia, W., Li, L. Song, Z., Xu, C., Liu. F.  Risk assessment of electric vehicle supply chain based on fuzzy synthetic evaluation. Energy 2019, 182, 397-411.

Round 2

Reviewer 2 Report

The Authors have addressed all the comments adequately and updated the paper throughly considering the comments.

Reviewer 3 Report

Author has improved the paper.

Reviewer 4 Report

I am satisfied with the authors' responses to my comments and the new changes undertaken in the manuscript.